# Lung Transplantation in Patients with Previous or Unknown Oncological Disease: Evaluation of Short- and Long-Term Outcomes

**DOI:** 10.3390/cancers16030538

**Published:** 2024-01-26

**Authors:** Chiara Catelli, Eleonora Faccioli, Stefano Silvestrin, Giulia Lorenzoni, Luca Luzzi, David Bennett, Marco Schiavon, Alessio Campisi, Elena Bargagli, Andrea Dell’Amore, Federico Rea

**Affiliations:** 1Thoracic Surgery Unit, Department of Cardiac, Thoracic, Vascular Sciences and Public Health, University of Padova, 35121 Padova, Italy; chiara.catelli1992@gmail.com (C.C.); stefano.silvestrin@aopd.veneto.it (S.S.); marco.schiavon@unipd.it (M.S.); andrea.dellamore@unipd.it (A.D.); federico.rea@unipd.it (F.R.); 2Unit of Biostatistics Epidemiology and Public Health, Department of Cardiac, Thoracic, Vascular Sciences and Public Health, University of Padova, 35121 Padova, Italy; giulia.lorenzoni@unipd.it; 3Lung Transplant Unit, University of Siena, Azienda Ospedaliero-Universitaria Senese, 53100 Siena, Italy; luca.luzzi@unisi.it; 4Respiratory Disease and Lung Transplant Unit, University of Siena, Azienda Ospedaliero-Universitaria Senese, 53100 Siena, Italy; david.bennett@ao-siena.toscana.it (D.B.); bargagli2@unisi.it (E.B.); 5Department of Thoracic Surgery, University and Hospital Trust-Borgo Trento, 37126 Verona, Italy; campisi.alessio88@gmail.com

**Keywords:** cancer, lung transplantation, survival, outcomes

## Abstract

**Simple Summary:**

Lung transplantation (LTX) is the treatment of choice for patients with end-stage lung disease but its role is still controversial in those with a history of malignancies. The aim of this study was to evaluate short- and long-term outcomes in patients submitted to LTX with a history of previous neoplasia or oncological disease detected in the native lung. Our study showed that this population had worse overall survival compared to a control group, emphasizing the importance of an accurate selection and a strict post-operative follow-up in this group of patients.

**Abstract:**

The accurate selection of the recipient is a crucial aspect in the field of lung transplantation (LTX), especially if patients were previously affected by oncological disease. The aim of this bicentric retrospective study was to evaluate short- and long-term outcomes in patients with previous oncological disease or unknown neoplasia found on native lungs submitted to LTX, compared to a control group. A total of 433 patients were included in the analysis, 31 with malignancies (Group 1) and 402 without neoplastic disease (Group 2). The two groups were compared in terms of short- and long-term outcomes. Patients in Group 1 were older (median age 58 years vs. 50 years, *p* = 0.039) and mostly affected by idiopathic pulmonary fibrosis (55% vs. 40% *p* = 0.002). Even though in Group 1 a lower rate of late post-operative complications was found (23% vs. 45%, *p* = 0.018), the median overall survival (OS) was lower compared to the control group (10 months vs. 29 months, *p* = 0.015). LTX represents a viable therapeutic option for patients with end-stage lung disease and a history of neoplastic disease. However, every case should be carefully debated in a multidisciplinary setting, considering oncological (histology, stage, and proper disease free-interval) and clinical factors (patient’s age and comorbidities). A scrupulous post-transplant follow-up is especially mandatory in those cases.

## 1. Introduction

Lung transplantation (LTX) is a valid therapeutic option for patients with end-stage lung disease refractory to medical treatment. Survival after LTX has progressively increased thanks to improvement in donors’ and recipients’ selection and advancements in organ preservation, surgical technique, and peri- and post-operative management [1]. To date, the median overall survival after the first year reaches 8.9 years for primary LTX [1], while it decreases to less than 7 years in the case of lung re-transplantation [2]. Historically, organ transplantation was reserved for patients without concurrent or irreversible conditions that could limit survival or make subsequent immunosuppressive therapy prohibitive [3], and, for this reason, the indication of LTX in patients with a history of oncological disease remains a controversial issue. The main concern is that LTX may increase the risk of tumor recurrence due to the need for ongoing immunosuppressive therapy to prevent graft rejection, reducing the long-term survival after LTX [4]. The introduction of the lung allocation score (LAS) has resulted in an increase in the average age of patients undergoing LTX, and, consequently, an increase in the diagnosis of pre-LTX neoplasia [5]. Despite these current uncertainties, several studies have shown that LTX can be a valid therapeutic option even for patients with a history of oncological disease, as long as specific selection criteria are met and a careful post-transplant monitoring is performed [6,7,8]. The rate of neoplastic recurrence after LTX ranges from 0 to 57%, with the majority reported within the first 5 years post LTX [9]. For transplantation listing, a disease-free interval of five years is currently accepted for the majority of patients with a previous malignant condition [10]. A different population of neoplastic patients is composed of those with cancer incidentally detected in the native lung after LTX. Despite a careful evaluation before LTX, several studies have observed that the detection of lung tumors in the native lung at the time of LTX is not uncommon [11,12,13]. The risk factors for end-stage lung disease and lung cancer are often similar, such as cigarette smoking. Additionally, in lung disease such as idiopathic pulmonary fibrosis (IPF) and sarcoidosis, the inflammatory stimulus can lead to an increased risk of undiagnosed neoplasia before LTX. Often, radiological images of patients with end-stage lung disease (especially those with IPF) make the differentiation between neoplastic lung nodules and the confluence of fibrotic foci difficult as a result of the progression of interstitial fibrotic process [14,15,16]. In these situations, a pre-operative biopsy is often burdened with a high level of difficulty, along with a high risk of complications, which can lead to an increase in morbidity and mortality [17]. To date, no specific guidelines for the clinical management of patients undergoing lung transplantation with detected malignant tumors in the explanted lungs are available [18,19,20]. The purpose of our retrospective bicentric study is to analyze short- and long-term outcomes in patients submitted to LTX with a previous history of neoplastic disease or with occult cancer found during the histopathological analysis of the native lung, comparing them with a control group.

## 2. Materials and Methods

### 2.1. Study Population

A total of 433 patients submitted to LTX at the Thoracic Surgery Unit of the University Hospital of Padua and at the Lung Transplant Unit of the University Hospital of Siena between 2006 and 2023 were included in this study. Among this population, 31 patients (Group 1) were affected by previous neoplastic disease or incidentally detected cancer in the native lungs, while 402 patients (Group 2) had no history of previous or occult neoplasia. Patients submitted to lung re-transplantation were excluded. Short- and long-term outcomes were evaluated and compared between the two groups. The study was approved by the Institutional Review Board (IRB) of the University Hospital of Padua (4539/AO/18).

### 2.2. Pre-Operative Evaluation

The selection of lung transplant recipients is a standardized process at both centers, including respiratory (global spirometry and CO_2_ diffusion, ventilation/perfusion scintigraphy, and 6 min walking test) and cardiovascular (ECG, echocardiogram, ultrasound of the supra-aortic trunks, cardiac catheterization, and coronary angiography) functional tests, and radiological/endoscopic evaluations (whole body CT scan and bronchoscopy). Blood and microbiological tests, bone and mineral metabolism evaluation, physiatrist, psychological, and nutritional counseling are always also performed. All the patients included in Group 1 with history of previous neoplasia had a disease-free interval, defined as no recurrence or metastasis from the primary tumor, equal to or greater than 5 years before being listed for LTX. The evaluation of the absence of neoplastic disease recurrence was performed through close oncological and radiological monitoring using contrast-enhanced whole-body CT and/or PET-CT scans every 6 or 12 months, depending on the underlying oncological disease.

### 2.3. Post-Operative Management

After LTX, triple immunosuppressive therapy based on corticosteroids, mycophenolate mofetil, and tacrolimus or ciclosporin was started. This therapy was maintained for life, with dose adjustments based on the periodic monitoring of plasmatic levels and on surveillance trans-bronchial biopsies (performed at 1 and 3 months after LTX, then every 3 months during the first year, and finally every year). Neoplastic recurrence after LTX was defined as radiological (CT or PET-CT) or histological evidence of the relapse.

### 2.4. Statistical Analysis

Descriptive statistics were reported as median (I-III quartiles) for continuous variables, and absolute numbers (percentages) for categorical variables. Wilcoxon–Kruskal–Wallis test and Pearson Chi-squared test, or Fisher’s exact test where appropriate, were performed to compare the distribution of continuous and categorical variables, respectively. A propensity score weighting approach was employed to account for potential confounding related to previous cancer history. Propensity scores were estimated using covariate balancing propensity score (CBPS), and a trimming of the weights was performed at the 90° quantile. Propensity scores were estimated considering age and gender. Covariate balance was evaluated using standardized mean differences (Figure 1).

A weighted logistic regression approach was adopted for binary outcomes. Results were reported as odds ratio (OR), 95% confidence interval (CI), and *p*-value. A Gamma model was employed for continuous outcomes, given the non-normal distribution of all the continuous outcomes considered. The marginal effect was computed considering the partial derivatives of the marginal expectation. Results were reported as average marginal effect (AME), 95% CI, and *p*-value. Weighted Cox proportional hazard models were employed for time-to-event outcomes. Results were reported as hazard ratio (HR), 95% CI, and *p*-value.

## 3. Results

### 3.1. Patients’ Characteristics 

Patients’ characteristics of both groups are reported in Table 1. No significant differences were observed between the two groups in terms of gender distribution (males 58% vs. 61%, *p* = 0.7). Patients included in Group 1 were significantly older than those in Group 2 (58 years vs. 50 years, *p* = 0.039). Regarding the underlying lung disease, a significant difference in the distribution was observed: Group 1 had a higher prevalence of IPF compared to Group 2 (55% vs. 40%) and other interstitial diseases (26% vs. 14%), while no cases of cystic fibrosis and idiopathic pulmonary arterial hypertension (IPAH) were reported in Group 1 (*p* = 0.002).

Concerning the previous neoplasia found in Group 1, the majority (*n* = 8, 25.8%) were lymphoproliferative diseases (three patients affected by Hodgkin’s lymphoma, two with acute myeloid leukemia, two with acute lymphoblastic leukemia, and one with Langerhans cell histiocytosis), followed by colon cancer (*n* = 5, 16.2%), and lung cancer (*n* = 3, 9.6%; one adenocarcinoma, one squamous cell carcinoma, and one small-cell lung cancer). Ten patients (32.3%) had an incidental detection of malignancy in the native lung (seven invasive lung adenocarcinomas, two large-cell lung carcinoma, and one invasive squamocellular carcinoma). Both patients with large-cell carcinoma showed lymph node involvement, with N1 in one case and N2 in the other, and underwent chemotherapy after LTX. The remaining eight patients were enrolled in a strict follow-up through serial total body CT scans. A recurrence of the neoplastic disease after LTX was reported in five patients (16.1%). Of these, four had an occasional detection of neoplasms in the native lung while one had early-stage pre-LTX colon cancer. The patient with the previous colon cancer had a local recurrence, while among the patients with unknown lung cancer, three experienced local intra-thoracic (lung and mediastinal lymph nodes) and one distant recurrence (liver). When the diagnosis of recurrence was confirmed, four patients underwent chemotherapy and one patient received concurrent chemo and radiotherapy. The characteristics of the underlying neoplastic disease, the treatment, and the recurrence rate are summarized in Table 2, while in Table 3, the characteristics of patients with occult lung cancer are reported.

### 3.2. Outcome Analysis

Table 4 summarizes the main intra- and post-operative outcomes. No intraoperative mortality was reported in either group; there were no significant differences in terms of intraoperative complications (16% and 21% in Group 1 and Group 2, respectively, *p* = 0.543, HR 0.74 95% CI 0.29–1.93), PGD (any grade) at 72 h (42% and 39% in Group 1 and Group 2, respectively, *p* = 0.777, OR 1.11 95% CI 0.54–2.27), or early post-operative complications (52% and 51% in Group 1 and 2, respectively, *p* = 0.967, OR 1.01 95% CI 0.50–2.04). However, a significant difference was observed in the late post-operative complication rate, which was lower in Group 1 (23% vs. 45%, *p* = 0.018, OR 0.36 95% CI 0.16–0.84). Regarding the median length of intensive care unit (ICU) (7 days in Group 1 and 9 days in Group 2, *p* = 0.136) and in-hospital stay (36 days in Group 1 and 31 days in Group 2, *p* = 0.885), no significant differences were observed between the two groups.

### 3.3. Survival Analysis

Figure 2 presents the weighted Kaplan–Meier. Median overall survival was lower in Group 1 compared to the control group (10 months vs. 29 months, *p* = 0.015). The values of 1-, 3- and 5-year survival were, respectively, 48% (95% CI: 33–71%), 37% (95%CI: 22–60%), and 31% (95% CI: 17–56%) in Group 1, and 75% (95% CI: 70–79%), 55% (95%CI: 51–61%), and 46% (95% CI: 41–52%) in Group 2.

Median OS was also compared between the two groups, excluding the ten patients with cancer detected on the native lungs, and it was still lower in Group 1, although without reaching statistical significance (15 months vs. 29 months, *p* = 0.912). 

## 4. Discussion

Lung transplantation currently represents the ultimate treatment for end-stage lung diseases refractory to medical therapy. In recent years, the adoption of advanced medical therapies, such as antifibrotic drugs [21], and the introduction of LAS has led to a progressive increase in recipients’ age; consequently, a previous history of neoplasia or an occult cancer in the native lung may represent frequent occurrences.

As expected, in our study, overall survival (OS) was lower in patients with a previous or occult oncological disease compared to the control group, and this could be explained by the older age, the cancer recurrence rate after LTX (which in our study was 16.1%), and the high rate of patients affected by IPF (55%) and other interstitial lung diseases (26%). As already reported [22], these underlying diseases have the worst prognosis after LTX, and this could have influenced the OS in Group 1.

Other available studies [11,23,24] have analyzed the outcomes of patients affected by previous neoplasia undergoing solid organ transplantation, including liver, kidney, heart, and lung. To the best of our knowledge, our study is the first in the current literature which investigates short- and long-term outcomes simultaneously in patients with previous or occult neoplastic disease submitted to LTX. Therefore, it is not possible to directly compare the outcomes of our analysis to those obtained from the other available studies.

Beaty et al. [5] analyzed outcomes after thoracic transplantations (heart and lung) in patients affected by pre-transplant malignancies: they found that 5% of patients undergoing LTX had a history of previous neoplasia. As in our analysis, this group consisted of older patients mostly affected by IPF, but, despite this, the authors did not reveal an increased mortality in patients with previous neoplasia. Reduced survival was observed only in patients with hematological diseases undergoing heart transplantation, but this datum was not confirmed for LTX. In our group of neoplastic patients, more than 25% had previous hematological malignancies (lymphoma or leukemia) compared to less than 1% in the study from Beaty et al.; considering the increased risk of relapse in these diseases and the need for prolonged chemotherapy treatments and/or bone marrow transplantation, this could have reduced the survival in our analysis. 

In 2017, Acuna et al. [23], in a meta-analysis, analyzed the outcomes of patients with a history of oncological disease undergoing solid organ transplantation. A total of 39 studies were included, revealing that cancer-related and non-cancer-related mortality were found to be higher in the group of patients with previous neoplasia. However, it is still unclear whether the increased risk of mortality is solely attributable to cancer-related mortality or to other factors. 

In a more recent study, Acuna et al. [24] observed that recipients with a prolonged interval between neoplasm onset and transplant (>5 years) had an increased risk of non-cancer-related death, attributing the increased mortality to the higher risk of developing renal insufficiency and cardiovascular complications in these patients. In our study, no significant differences emerged between the two groups in terms of early complications, while a significant difference was observed in late complications, with an unexpected lower rate in the neoplastic subgroup. A possible explanation is that in our neoplastic population, no recipients were affected by cystic fibrosis (CF) or idiopathic pulmonary hypertension (IPAH) and, as known, CF patients are generally at a higher risk of developing post-operative infections, as they are frequently colonized by multidrug-resistant microorganisms, which can produce complications even a long time after LTX [22]. At the same time, patients with IPAH are frequently affected by long-term cardiovascular complications post-operatively. Brattstrom et al. [25] observed how the rate of recurrence of oncological disease after transplantation was higher in those patients with a latency between neoplastic healing and transplantation of less than 5 years; this is consistent with what we observed in our study. Patients with an occasional detection of cancer on the explanted lung were indeed the only ones with a latency of less than 5 years; in this subgroup, recurrence occurred in 4 out of 10 patients (40%), compared to only 1 patient (4.7%) in the subgroup of patients with a disease-free interval longer than 5 years. 

Regarding the population with cancer incidentally found in the native lung, in our study, 10 cases were included and 90% of those had an underlying IPF. Considering our entire cohort of 433 patients, 177 had IPF and among these 9 (5.1%) had an unknown lung neoplasia at the time of transplantation. This result is comparable to those reported by Song et al. [26] and Hubbard et al. [27], who described a frequency of lung cancer in patients with IPF of 6.4% and 4.4%, respectively. The higher frequency in IPF patients may be due to the fact that the lung affected by end-stage pulmonary fibrosis frequently presents misleading radiological images, which do not often allow for a simple differential diagnosis with neoplastic nodules. Furthermore, patients with pulmonary fibrosis are often older and ex-smokers, increasing the risk of developing neoplasms, even if in our study, the number of packs per year was not known for each patient and the difference was not analyzed between the two groups.

Razia et al. [19] and Choi et al. [20] compared long-term survival between patients with incidentally detected malignancies in native lungs and those without, reporting, as expected, lower survival rates in the first subgroup of patients. Although this analysis was not performed in our study, it seems that the rate of cancer-related deaths is higher in this subgroup of patients compared to those with previous neoplasia, as demonstrated by the higher rate of recurrence (40% vs. 4.7%)

The limitations of this study are as follows: firstly, this is a retrospective study with a relatively small sample size, despite being bicentric. Furthermore, patients with previous neoplasia and patients with the occasional finding of lung cancer in the explanted lung constitute two different populations, with more unfavorable outcomes expected in the latter group. However, the small size of these two subgroups did not allow to create two distinct populations for the analysis. Further studies, preferably multicentric, are necessary to validate our findings. 

## 5. Conclusions

Lung transplantation can also be considered a valid therapeutic option in patients affected by previous neoplasia; a proper disease-free interval needs to be demonstrated, every case should be carefully debated in a multidisciplinary setting before listing, and a scrupulous post-transplant follow-up must be performed in order to improve short- and long-term outcomes in this population. Additionally, especially in IPF patients, an accurate pre-operative evaluation is always necessary to reduce as much as possible the detection of unexpected cancer in the native lung after LTX.

## Figures and Tables

**Figure 1 cancers-16-00538-f001:**
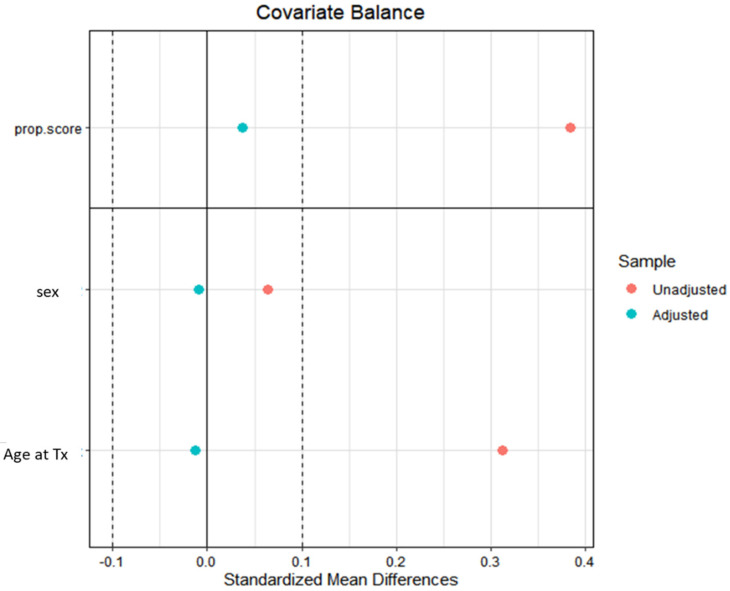
The standardized mean differences (SMD) before (red dots) and after (blue dots) the propensity score weighting procedure for the variables used in propensity score estimation (age and gender). Tx: transplantation.

**Figure 2 cancers-16-00538-f002:**
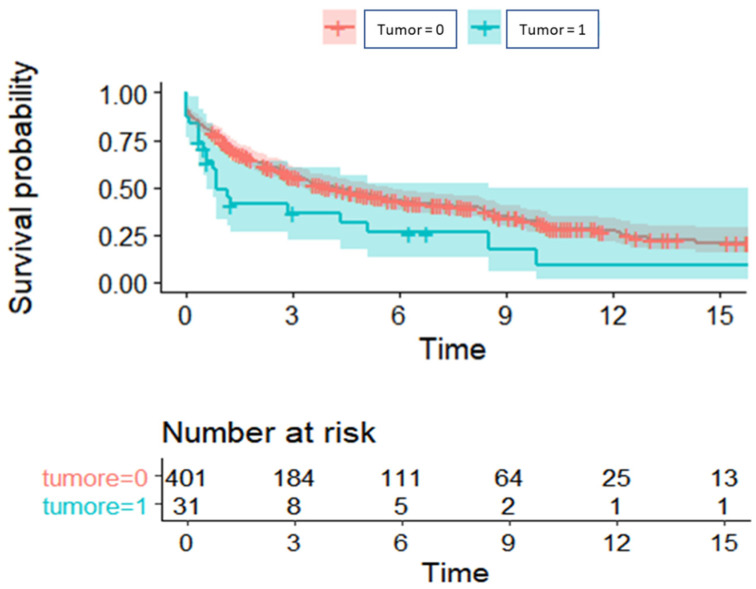
Overall survival of patients with previous or occult neoplasia (turquoise) compared to those without a history of neoplasia (red).

**Table 1 cancers-16-00538-t001:** Characteristics of the overall enrolled patients, divided into Group 1 (patients with previous or occult neoplastic disease) and Group 2 (patients without neoplastic disease).

Variable	Group 1 N = 31	Group 2 N = 402	*p*-Value
Gender			0.7
M	18 (58%)	246 (61%)	
F	13 (42%)	156 (39%)	
Age at LTX (y)	58 (IQR, 41–62)	50 (IQR, 34−39)	0.039
Pulmonary Disease			0.002
IPAH	0 (0%)	7 (1.7%)	
CF	0 (0%)	109 (27%)	
IPF	17 (55%)	160 (40%)	
COPD/emphysema	4 (13%)	55 (14%)	
Bronchiectasis	2 (6.5%)	13 (3.2%)	
Other ^a^	8 (26%)	59 (14%)	
Type of LTX			0.8
SLTX	27 (81%)	339 (84%)	
BLTX	4 (13%)	63 (16%)	

^a^ Other end-stage pulmonary diseases included lymphangioleiomiomatosis (LAM), histiocytosis-X, graft-versus-host disease (GVHD), extrinsic allergic alveolitis, sarcoidosis, hemosiderosis, and scleroderma. The data are shown as absolute number (with percentage) for categorical variables and as median (with interquartile range) for continuous variables. IQR: interquartile range; M: males; F: females; IPAH: idiopathic pulmonary arterial hypertension; CF: cystic fibrosis; IPF: idiopathic pulmonary fibrosis; COPD: chronic obstructive pulmonary disease; LTX: lung transplant; SLTX: single-lung transplant; BLTX: bilateral lung transplant, y: years.

**Table 2 cancers-16-00538-t002:** Characteristics of the primary tumor, treatment, and recurrence of patients in Group 1.

	N (%)
Primary tumor	
Lymphoproliferative disease	8 (25.8%)
Skin non-melanoma tumor	1 (3.2%)
Lung cancer	3 (9.6%)
Occult lung cancer (diagnosis on native lung)	10 (32.3%)
Head and neck cancer	2 (6.5%)
Colon cancer	5 (16.2%)
Uterine cancer	1 (3.2%)
Prostatic cancer	1 (3.2%)
Oncological Treatment	
Surgery	11 (35.5%)
Surgery + CT	1 (3.2%)
Surgery + CT + RT	2 (6.5%)
CT	2 (6.5%)
CT + RT	1 (3.2%)
CT + RT + bone marrow transplant	2 (6.5%)
Bone marrow transplant	3 (9.6%)
Follow-up	9 (29.0%)
Cancer recurrence	5 (16.1%)

CT: chemotherapy; RT: radiotherapy.

**Table 3 cancers-16-00538-t003:** Characteristics of patients with occult lung cancer.

Patient	Age at LTX	End-Stage Lung Disease	Lung Cancer Histology	N Stage Disease	Therapy Post-LTX	Recurrence	Status	OS (Months)
Patient 1	61	IPF	ADK	0	No	No	Alive	14
Patient 2	58	IPF	ADK	0	No	No	Dead	36
Patient 3	62	IPF	ADK	0	No	No	Dead	0
Patient 4	59	IPF	ADK	0	No	No	Dead	9
Patient 5	47	IPF	ADK	0	No	No	Dead	16
Patient 6	42	LAM	LCC	1	CT + RT	No	Dead	5
Patient 7	52	IPF	SCC	0	No	Liver	Dead	10
Patient 8	65	IPF	ADK	0	No	Lung	Dead	106
Patient 9	63	IPF	LCC	2	CT	Mediastinal LN	Dead	7
Patient 10	56	IPF	ADK	0	No	Lung	Dead	10

ADK: adenocarcinoma; CT: chemotherapy; IPF: idiopathic pulmonary fibrosis; LAM: lymphangioleiomyomatosis; LCC: large-cell carcinoma; LN: lymph nodes; RT: radiotherapy; SCC: squamous cell carcinoma; LTX: lung transplantation.

**Table 4 cancers-16-00538-t004:** Results of outcome analysis.

Outcome	Group 1 N = 31	Group 2 N = 402	OR (95% CI)	*p*-Value
Intra-operative complications	16%	21%	0.74 (0.29–1.93)	0.543
PGD (any grade) at 72 h	42%	39%	1.11 (0.54–2.27)	0.777
Early post-operative complications (within 30 days from LTX)	52%	51%	1.01(0.50–2.04)	0.967
Late post-operative complications (over 30 days from LTX)	23%	45%	0.36 (0.16–0.84)	0.018
	**Group 1** **N = 31**	**Group 2** **N = 402**	**AME** **(95% CI)**	***p*-Value**
ICU stay (d)	7 (IQR, 5–11)	9 (IQR, 5–18)	−4.442 (−10.286–1.403)	0.136
In-hospital stay (d)	36 (IQR, 25–40)	31 (IQR, 25–40)	2.287 (−28.775–33.350)	0.885

Outcome distribution is shown in the two groups, presenting percentages for binary outcomes and median (IQR) for continuous outcomes. The third column represents the results of the weighted regression models used to assess the effect of the neoplasia on the outcomes. For binary outcomes, ORs within the lower and upper bound of the 95%CI are presented. For continuous outcomes, AME within the lower and upper bound of 95% CI are presented. LTX: lung transplantation; ICU: intensive care unit; MV: mechanical ventilation; PGD: primary graft dysfunction; CI: confidence interval; AME: average marginal effect.

## Data Availability

Data are contained within the article.

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
