# Peer review of "Lung Transplantation in Patients with Previous or Unknown Oncological Disease: Evaluation of Short- and Long-Term Outcomes"

_cancers, 2024, doi:10.3390/cancers16030538_

Round 1

Reviewer 1 Report

Comments and Suggestions for Authors

Thank you very much for conducting this important study.

I have several comments to improve the manuscript.

-Firstly, please give the details regarding the frequency of the incidental lung cancer during lung transplantation for IPF from the all cohort and discuss this rate with the previous data from the literature. The autopsy studies studies reported over 40% lung cancer in patients with IPF (DOI: 10.1038/s41598-021-82182-8; DOI: 10.1164/ajrccm.161.1.9906062). 

-Please give details regarding the lung cancers detected during the transplantation. Whether these cancers were originated from the areas affected from the IPF should be added. Preferable a table summarizing these cancer cases would be beneficial for the further studies.

-Please add details regarding the comorbidity burden across two groups. While the age was different between the groups, the presence of cardiac comorbidities and diabetes could also be different and could be serve as a confounder for the survival difference across two studies.

-Please give an additional survival analysis after excluding the patients with incidental cancer detected during the transplantation. This analysis could highlight the potential risks of cancer recurrence in patients with a remote history of cancer. If the survival was similar after the exclusion of these patients, we should infer a need for better lung cancer screening before transplantation.

-A last important point would be that 5/8 of the patients with lymphoproliferative disease had bone marrow transplant. This point a possibility of high-risk lymphoproliferative diseases needing a transplant in most patients. I recommend the authors to give more details regarding the types of lymphoproliferative diseases in details instead of grouping them all together.

-Please add smoking pack-years for two groups. If this data is not available, please discuss it as a limitation.

Comments on the Quality of English Language

The quality of English is acceptable.

Author Response

Firstly, please give the details regarding the frequency of the incidental lung cancer during lung transplantation for IPF from the all cohort and discuss this rate with the previous data from the literature. The autopsy studies studies reported over 40% lung cancer in patients with IPF (DOI: 10.1038/s41598-021-82182-8; DOI: 10.1164/ajrccm.161.1.9906062). 

We appreciate this comment made by the reviewer. We have calculated the incidence of occult lung cancer in patients with IPF and it seems to be comparable to what was reported in other studies (references 26-27 have been included). Please see lines 299-303  in the discussion.

Please give details regarding the lung cancers detected during the transplantation. Whether these cancers were originated from the areas affected from the IPF should be added. Preferable a table summarizing these cancer cases would be beneficial for the further studies.

We thank the reviewer for this comment. As suggested, we have added a new table (Table 3) which summarizes the characteristics of each patient with occult lung neoplasia. Since the aim of our study was focused on clinical aspects of these patients, although extremely important, we do not have analyzed pathological data. According to the existing literature, we know that lung cancer in IPF patients is typically observed in the peripheral zone of the lower lung in the fibrotic area. A detailed pathological analysis of this peculiar population is under evaluation and will be the main topic of our next studies on this population.

Please add details regarding the comorbidity burden across two groups. While the age was different between the groups, the presence of cardiac comorbidities and diabetes could also be different and could be serve as a confounder for the survival difference across two studies.

We thank the reviewer for the possibility to point out this aspect. We can assume that older patients had a higher rate of pre-transplant comorbidities and this could have affected survival, but unfortunately this aspect was not analyzed by our study. As specified in the discussion the older age of group 1 may have affected the overall survival in this subgroup.

Please give an additional survival analysis after excluding the patients with incidental cancer detected during the transplantation. This analysis could highlight the potential risks of cancer recurrence in patients with a remote history of cancer. If the survival was similar after the exclusion of these patients, we should infer a need for better lung cancer screening before transplantation.

We really appreciate this comment. We have performed an additional analysis excluding patients with occult neoplasia incidentally found on the native lung. Please see lines 240-242 in the results

-A last important point would be that 5/8 of the patients with lymphoproliferative disease had bone marrow transplant. This point a possibility of high-risk lymphoproliferative diseases needing a transplant in most patients. I recommend the authors to give more details regarding the types of lymphoproliferative diseases in details instead of grouping them all together.

We appreciate this comment. We have added more details on the types of lymphoproliferative disease (please see lines 151-153)

-Please add smoking pack-years for two groups. If this data is not available, please discuss it as a limitation.

This is an important aspect. Unfortunately, this datum was available only for a very small proportion of patients and for this reason it was not reported and analyzed. We have added this aspect (lines 307-309)

Reviewer 2 Report

Comments and Suggestions for Authors

This manuscript described the differences of outcomes of patients receiving lung transplantation (LTX) with or without previous history of oncological diseases. The results of this study would provide special interest to journal readers. However, in my opinion, this manuscript had some problematic points.

Major points.

1)      The prognosis of patients receiving LTS would be influenced by the primary reason (pulmonary endostage diseases). The detailed findings of such disease should be described. What were “Others”? (page 4, Table 1), The differences of these primary lung diseases between two groups can contribute to the patients’ prognoses?

2)      The authors should describe the detailed findings of previous history of neoplastic diseases. What was lymphoproliferative diseases? Malignant lymphoma? T cell lymphoma? Or Leukemia? Myelodysplastic syndrome? What type of lung cancer? Squamous cell carcinoma? Adenocarcinoma?

3)      I believe that the presence of occult lung cancer in removed native lung is interesting because of not low incidence (approximately one-third of group 1). The authors also discussed this concern. However, the detail findings of these incidental cancers should be described, such as cancer histology (squamous cell carcinoma, adenocarcinoma, tumorlet,  ..), size, the presence of spreading/metastases (limited diseases or not), …. 

4)      This manuscript should be edited by native English scientist.

Minor points:

1.      No spaces between words and parentheses: 13(42%) [page 4, Table 1].

2.      “table 1” (page 4, line 143); Table 1 (page 4)

3.      Explanation of tables and the text: confusing (pages 4, 6).

4.      Ten patients (32.3%) had incidentally detection …., not “10 patients (32.3%) had incidentally …” (page 4, line 152).

5.      LTX and LTx (page 8, line 270).

Comments on the Quality of English Language

 This manuscript should be edited by native English scientist.

Author Response

This manuscript described the differences of outcomes of patients receiving lung transplantation (LTX) with or without previous history of oncological diseases. The results of this study would provide special interest to journal readers. However, in my opinion, this manuscript had some problematic points.

Major points.

  • The prognosis of patients receiving LTS would be influenced by the primary reason (pulmonary end-ostage diseases). The detailed findings of such disease should be described. What were “Others”? (page 4, Table 1), The differences of these primary lung diseases between two groups can contribute to the patients’ prognoses? 

We thank the reviewer for the possibility to better point out this aspect. In the footnote of table 1 (lines 175-177) we have specified the other underlying native lung diseases and the contribution of these to patients’ prognosis (lines 253-255).

  • The authors should describe the detailed findings of previous history of neoplastic diseases. What was lymphoproliferative diseases? Malignant lymphoma? T cell lymphoma? Or Leukemia? Myelodysplastic syndrome? What type of lung cancer? Squamous cell carcinoma? Adenocarcinoma?

We appreciate this comment. We have added information regarding the previous lymphoproliferative disease and the type of lung cancer (please see lines 150-154 in the results).

  • I believe that the presence of occult lung cancer in removed native lung is interesting because of not low incidence (approximately one-third of group 1). The authors also discussed this concern. However, the detail findings of these incidental cancers should be described, such as cancer histology (squamous cell carcinoma, adenocarcinoma, tumorlet,  ..), size, the presence of spreading/metastases (limited diseases or not).

We thank the reviewer for this suggestion. We have added the details requested in an additional table (please see Table 3). Unfortunately, since the aim of our study was the comparisons between the two groups in term of clinical outcomes, a detailed analysis of pathological data was not performed, for this reason information regarding tumor size were not available.

  • This manuscript should be edited by native English scientist. 

The English language was revised as suggested.

Minor points:

  1. No spaces between words and parentheses: 13(42%) [page 4, Table 1].
  2. “table 1” (page 4, line 143); Table 1 (page 4)
  3. Explanation of tables and the text: confusing (pages 4, 6).
  4. Ten patients (32.3%) had incidentally detection …., not “10 patients (32.3%) had incidentally …” (page 4, line 152).
  5. LTX and LTx (page 8, line 270).

We thank the reviewer. We have made all the requested corrections